# Pulmonary Vein Isolation Outcome Degree Is a New Score for Efficacy of Atrial Fibrillation Catheter Ablation

**DOI:** 10.3390/jcm10245827

**Published:** 2021-12-13

**Authors:** Ruzica Jurcevic, Lazar Angelkov, Nebojsa Tasic, Milosav Tomovic, Dejan Kojic, Petar Otasevic, Milovan Bojic

**Affiliations:** Department of Electrophysiology, Institute for Cardiovascular Diseases Dedinje, Heroja Milana Tepica 1, 11000 Belgrade, Serbia; langelkov@yahoo.com (L.A.); nebtasa@yahoo.com (N.T.); milosav@yubc.net (M.T.); kojicdrd@yahoo.com (D.K.); potasevic@yahoo.com (P.O.); ruzicajurcevic67@gmail.com (M.B.)

**Keywords:** atrial fibrillation, left atrial diameter, left ventricular ejection fraction, CHA_2_DS_2_-VASc score, pulmonary vein isolation

## Abstract

This study introduces the pulmonary vein isolation outcome degree (PVIOD) as a new semiquantitative measure for the efficacy of atrial fibrillation (AF) catheter ablation and reports the determination of predictors associated with PVIOD. The median follow-up periods of 117 patients after the first and last ablation were, respectively, 82 (IQR 15) and 72 (IQR 30) months. PVIOD 1 included 32.5% of patients, those with successful single pulmonary vein isolation (PVI); PVIOD 2 included 29.1% of subjects, those with success after multiple procedures; PVIOD 3 comprised 14.5% of patients, those with clinical success; and PVIOD 4 included 23.9% of cases, those with procedural and clinical failure. In the multivariate ordinal logistic regression analysis, PVIOD 1–4 were independently associated with longstanding persistent AF with paroxysmal AF as the referent category (odds ratio (OR), 3.5; 95% confidence interval (95% CI), 1.1–10.7 (*p* = 0.031)), left atrial (LA) diameter (OR, 1.2; 95% CI, 1.1–1.3 (*p* = 0.001)) and left ventricular ejection fraction (LVEF) (OR, 0.9; 95% CI, 0.86–1.0 (*p* = 0.038)). LA size > 41 mm, LVEF ≤ 50% and longstanding persistent AF are strong predictors of AF recurrence. PVIOD 1–4 offer the most exact long-term prognosis of PVI. The purpose of the present article is to expand the quantitative measure of procedural success in the medical and biological fields.

## 1. Introduction

Atrial fibrillation (AF) is the most common cardiac arrhythmia associated with increased morbidity and mortality [1]. Radiofrequency (RF)-based pulmonary vein isolation (PVI) is an established therapy for symptomatic, drug-refractory paroxysmal and persistent AF [2]. The story of PVI began two decades ago with modest and promising efficacy [3]. In the modern era, after multiple catheter ablations and additional substrate modification (ASM), long-term success has reached 62–79.5% in paroxysmal AF, 46.2–68.2% in persistent AF and 39–41% in longstanding persistent AF [2,4,5,6,7].

Many studies have explored the predictors of arrhythmia recurrence, such as nonparoxysmal AF; female sex; longer AF duration prior to the catheter ablation; sleep apnea; obesity; older age; hypertension; structural heart disease; enlarged left atrial (LA) diameter and low left ventricular ejection fraction (LVEF); high CHADS_2_ [8], CHA_2_DS_2_-VASc [9], MB-LATER and APPLE scores [10]; high C-reactive protein [11]; low LA voltage [12]; increased LA fibrosis detected by cardiac magnetic resonance imaging [13]; and longer PV durability [14]. Numerous trials, when evaluating the efficacy of PVI, have used a qualitative measure with two possible outcomes—a successful or an ineffective procedure [3,7]. Teunissen et al. suggested that long-term clinical success, together with procedural success, could be crucial after AF catheter ablation [2]. Several researchers have reported that AF type, LA size and LVEF were associated with ablation success rate but never provided quantified outcomes [2,3,4,5,6,7,8,9,10,11,12].

Our study aimed to evaluate pulmonary vein isolation outcome degree (PVIOD) as a new semiquantitative measure for PVI success after a 7-year follow-up and to determine parameters for predicting the PVIOD. PVIOD is an outcome in itself. We attempted to demonstrate that PVIOD 1–4 is the most exact prognosis of PVI that can be determined with clinical parameters. In addition, we hope that the method of quantitative evaluation of procedural success will be accepted in the future development of prognostic tools in medical and biological sciences.

## 2. Materials and Methods

### 2.1. Ablation Protocol

This prospective study included 124 consecutive patients with symptomatic AF who underwent PVI at the Institute for Cardiovascular Diseases, Dedinje, Serbia, from January 2012 to December 2013. AF was classified according to the 2012 HRS/EHRA/ECAS Consensus Statement on Catheter and Surgical Ablation of AF [11]. Paroxysmal or persistent AF was defined, respectively, if there was a success or failure to terminate AF spontaneously or with intervention within 7 days of onset. Longstanding persistent AF was diagnosed if arrhythmia lasted more than 1 year. The study excluded those who had permanent AF, pregnancy, acute reversible causes of AF, myocardial infarction that occurred within 3 months, moderate-to-severe valvular stenosis or regurgitation and coronary artery disease indicated for revascularization procedure.

The trial was approved by the Ethics Committee of the Institute for Cardiovascular Diseases, Dedinje. The work was conducted in accordance with the Declaration of Helsinki and all patients signed informed consent forms prior to their inclusion in the study. Initially, all subjects were evaluated for age, sex, AF duration, AF type, clinical symptoms and signs, CHA_2_DS_2_-VASc score, risk factors for cardiovascular disease (hypertension, hypercholesterolemia and diabetes mellitus) and the presence of structural heart disease. Patients’ medical documents, 12-lead electrocardiograms (ECG) and 7-day Holter monitoring of ECG were reviewed for the presence of AF. Transthoracic echocardiography was used to measure the LVEF and the end-systolic LA size through antero-posterior diameter from parasternal long-axis view. Transesophageal echocardiography, together with images from 64-multislice computed tomography of pulmonary veins (PVs), provided better visualization of anatomy of LA and PVs.

In all patients, the first PVI included only catheter ablation of PV antrum without additional substrate modification (ASM). After transseptal puncture, an intravenous bolus of heparin (5000 IU) was administered, followed by additional doses to maintain an activated clotting time between 300 and 350 s. During the procedure, the antrum of ipsilateral PVs was widely encircled with an irrigated tip catheter (ThermoCool Navi-Star, Biosense-Webster) with point-by-point ablation lesions using the CARTO system. The RF energy was applied at a target temperature of 43 °C with a power limit of 30–35 W for 30–60 s. Entrance block, exit block and noninducibility of AF confirmed the initial success of PVI. If isolation of PV was consistent for 30 min after the last RF application, we concluded that procedure had been successful. When typical atrial flutter (AFL) was documented, cavotricuspid isthmus was ablated.

All subjects received antiarrhythmic drugs (AAD) for 3 months after initial PVI, in the form of a daily oral dose of either 200 mg amiodarone or 150 mg propafenone three times a day, or 80 mg sotalol two times a day. After that period, AADs were discontinued if patients were free of arrhythmia recurrence. Rhythm status of AADs was reevaluated after 3 months. Oral anticoagulant was administered for 6 months after catheter ablation in all patients and continued after that in those with a CHA_2_DS_2_-VASc score ≥ 2.

### 2.2. Follow-Up

Postablation AF type and burden were determined 1, 3 and 6 months after catheter ablation and, later, by at least an annual clinical review. Subjects who could not be reached were excluded from the study. Recurrences of AF, AFL and atrial tachycardia (AT) were defined as episodes of arrhythmias lasting for more than 30 s. A repeat procedure was recommended to patients with arrhythmia recidivism after a 3-month blanking period. Re-PV ablations were performed in case of AF recurrences in patients with recovered PV conduction. In those subjects without PV reconnection, ablations of complex fractionated atrial electrograms (CFAE) and/or ASM were applied based on observations made during the electrophysiology study. ASM included LA roof line (connecting two superior PVs), mitral isthmus (mitral annulus to the left inferior PV), anterior line, inferior line, circumferential lines around superior vena cava and coronary sinus. The ablation endpoints were bidirectional block for linear ablation or AF termination during CFAE ablation.

We introduce a new semiquantitative method for measuring PVI success, i.e., the pulmonary vein isolation outcome degree (PVIOD), with four possible outcomes (by which we classified patients into four distinct groups) after a 7-year follow-up. The PVIOD 1 group included patients with successful single PVI, and the PVIOD 2 group included those with efficacy after multiple procedures (≥2 re-PV isolation and/or ASM). Patients with clinical success after PVI ± ASM regardless of procedure number were in the PVIOD 3 group. The PVIOD 4 group consisted of patients with procedural and clinical failure despite PVI ± ASM. Long-term procedural success after single and multiple ablations was defined as freedom from atrial arrhythmia recurrence following the 3-month blanking period, in the absence of Class I and III AADs. Seven-year clinical success was defined as a significant reduction in the number and duration of AF episodes or the % time a patient was in AF, with or without previously ineffective AAD therapy. Subjects without arrhythmia recidivism were included in PVIOD 1 + 2 as successful procedures, while PVIOD 3 + 4 included procedural failures. PVIOD 1 + 2 + 3 indicate procedural and clinical success, while PVIOD 4 indicates procedural and clinical failure.

The catheter ablation complications were divided into major and minor complications. A major ablation complication was defined as a serious, life-threatening condition, while a minor complication was not considered serious.

### 2.3. Statistical Analysis

Continuous variables are presented as the mean ± standard deviation or median and interquartile range, while categorical variables are shown as absolute values and percentages. The differences between continuous values were assessed using an unpaired Student’s t-test when the data were normally distributed, a Mann–Whitney test for skewed variables and a λ^2^ test or Fisher’s exact test for nominal variables. Ordinal logistic regression analysis was used to identify the predictors for PVIOD. The multivariate (MV) logistic regression analysis included variables with a statistical significance of 0.05 on the univariate (UV) logistic regression model. Nomogram, as a two-dimensional diagram, is used for graphical presentation of MV binary logistic regression model predictions. Receiver operating characteristic (ROC) curves were applied to determine the cut-off points for LA diameter, LVEF and CHA_2_DS_2_-VASc score, which discriminate between procedural success and procedural failure, as well as between procedural with clinical success (PVIOD 1 + 2 + 3) and procedural with clinical failure (PVIOD 4). The AF-free survival after catheter ablations was estimated using the product–limit (Kaplan–Meier) method. Kaplan–Meier curves were constructed for AF-free survival after PVI in patients divided into three groups according to the type of AF and two groups with different LA sizes, LVEF and CHA_2_DS_2_-VASc scores, using the nonparametric log-rank test. Significance was established at *p* < 0.05.

## 3. Results

### 3.1. Study Population

Initial PVI was performed in 124 patients; however, 7 subjects did not respond for evaluation and were excluded from this trial. The baseline and PVIOD characteristics of 117 study patients are shown in Table 1. PVIOD 1–4 were significantly different in AF type (*p* = 0.009), CHA_2_DS_2_-VASc score (*p* < 0.001), hypertension (*p* = 0.035), structural heart disease (*p* = 0.046), LA diameter (*p* < 0.001) and LVEF (*p* < 0.001). Structural heart disease was present in 21 subjects (17.9%): ischemic heart disease in 7 patients, dilated cardiomyopathy in 9, valvular heart disease in 4 and hypertrophic cardiomyopathy in 1 patient. During the initial PVI procedure, typical AFL was diagnosed in 8 subjects (6.8%) and cavotricuspid isthmus was successfully ablated in all of them.

Across the 117 study patients, we performed 209 catheter ablations (mean of 1.8 per patient). A total of 53 patients had one PVI, while 64 patients underwent redo ablations—41 had two procedures, 18 had three ablations and 5 had four procedures. The nonpulmonary vein triggers were ablated in eight patients, within which five were successful, one had clinical success and two were failures. Arrhythmia complications were found in eight patients (6.8%), of whom three had left-sided AT and five had atypical AFL. During follow-up, three patients died from noncardiac causes, which were not related to PVI.

### 3.2. Catheter Ablation Success after 7-Year Follow-Up

The median follow-up after the first and last ablation was, respectively, 82 (IQR 15) and 72 (IQR 30) months. PVIOD 1 included the 32.5% (*n* = 38) of patients with successful single PVI (Table 2). PVIOD 2 had the 29.1% (*n* = 34) of patients who underwent ≥ 2 successful catheter ablations—re-PVI in 24.8% (*n* = 29) cases and ASM in 4.3% (*n* = 5) cases. PVIOD 3 contained the 14.5% (*n* = 17) of subjects with clinical success and PVIOD 4 had the 23.9% (*n* = 28) of patients with procedural and clinical failure. Seven-year success after single PVI was 32.5%, of which 40.2% were in paroxysmal AF, 20.8% in persistent AF and 12.5% in longstanding persistent AF. Cumulative long-term PVI success after single and multiple procedures with ASM was 61.6% and cumulative procedural and clinical success was 76.1%.

### 3.3. Predictors for Unsuccessful Catheter Ablation of Pulmonary Vein

In the UV ordinal logistic regression analysis, PVIOD was independently associated with longstanding persistent AF, with paroxysmal AF as a referent category (odds ratio (OR), 6.5; 95% confidence interval (95% CI), 2.3–18.6 (*p* = 0.001)), LA diameter (OR, 1.2; 95% CI, 1.1–1.3 (*p* < 0.001)), LVEF (OR, 0.9; 95% CI, 0.85–0.95 (*p* < 0.001)), CHA_2_DS_2_-VASc score (OR, 1.5; 95% CI, 1.1–2.1 (*p* = 0.008)), structural heart disease (OR, 3.0; 95% CI, 1.2–7.2 (*p* = 0.014)) and diabetes mellitus (OR, 3.6; 95% CI, 1.0–13 (*p* = 0.049)) (Table 3). In the MV ordinal logistic regression model, PVIOD remained independently associated with longstanding persistent AF, with paroxysmal AF as the referent category (OR, 3.5; 95% CI, 1.1–10.7 (*p* = 0.031)), LA diameter (OR, 1.2; 95% CI, 1.1–1.3 (*p* = 0.001)) and LVEF (OR, 0.9; 95% CI, 0.86–1.0 (*p* = 0.038)).

Nomogram 1 was configured for four risk factors—type of AF, LA diameter, LVEF and CHA_2_DS_2_-VASc score—which affected the 7-year probability for procedural failure. PVIOD 1 + 2 was compared with PVIOD 3 + 4 (Figure 1A). Nomogram 2 shows the influence of the same risk factors on the 7-year probability for procedural and clinical failure. PVIOD 1 + 2 + 3 was compared with PVIOD 4 (Figure 1B). To use the nomogram, a line is drawn perpendicularly from the axis of each risk factor to the top line labeled “Points” and the number of points are then summarized. Next, a line is drawn from the axis called “Total Points” to the axis, which measures the probability for procedural failure (Figure 1A) and procedural with clinical failure (Figure 1B) after the 7-year follow-up.

By using the ROC curves, optimal cut-off values were calculated as an LA diameter of 41.5 mm, LVEF of 50% and CHA_2_DS_2_-VASc score of 1.5 (Appendix A). We compared procedural success with procedural failure; LA diameter’s ROC curve had a sensitivity of 67% and a specificity of 80%, with AUC of 0.726, 95% CI of 0.6–0.8 and *p* < 0.001 (Appendix A); LVEF’s ROC curve had a sensitivity of 53% and a specificity of 82%, with AUC of 0.712, 95% CI of 0.6–0.8 and *p* < 0.001 (Appendix A); CHA_2_DS_2_-VASc score’s ROC curve had a sensitivity of 58% and a specificity of 74%, with AUC of 0.711, 95% CI of 0.6–0.8 and *p* < 0.001 (Appendix A). In addition, we compared procedural and clinical success with procedural and clinical failure; LA diameter’s ROC curve had a sensitivity of 86% and a specificity of 60%, with AUC of 0.741, 95% CI of 0.6–0.8 and *p* < 0.001; LVEF’s ROC curve had a sensitivity of 54% and a specificity of 75%, with AUC of 0.681, 95% CI of 0.6–0.8 and *p* = 0.002; CHA_2_DS_2_-VASc score’s ROC curve had a sensitivity of 68% and a specificity of 71%, with AUC of 0.718, 95% CI of 0.6–0.8 and *p* < 0.001.

The Kaplan–Meier curve in Figure 2A demonstrates AF-free survival of our study patients after catheter ablations and 7-year follow-up (PVIOD 1 + 2 compared with PVIOD 3 + 4). The Kaplan–Meier curves depicted AF-free survival after PVI in patients divided into two groups with LA diameter ≤ 41mm and LA diameter > 41 mm with log-rank = 15.542, *p* < 0.001 (Figure 2B). The subjects with LA size ≤ 41 mm had significantly better long-term survival free of AF recidivism. The Kaplan–Meier curves showed arrhythmia-free survival after PVI in patients as follows: divided into two groups with LVEF ≤ 50% and LVEF > 50%, log-rank = 13.957, *p* < 0.001 (Figure 2C); divided into three groups according to AF type, log-rank = 6.427, *p* = 0.04 (Figure 2D); and divided into two groups with CHA_2_DS_2_-VASc score 0–1 point and CHA_2_DS_2_-VASc score 2–5 point, log-rank = 3.636, *p* = 0.057 (Figure 2E). We used a Kaplan–Meier curve to demonstrate AF-free survival after RF ablation of PV and long-term follow-up of our patients (PVIOD 1 + 2 + 3 compared with PVIOD 4) (Figure 3A). The Kaplan–Meier curves depicted AF-free survival after PVI in patients as follows: divided into two groups with LA diameter ≤ 41 mm and LA diameter > 41 mm with log-rank = 13.808, *p* < 0.001 (Figure 3B); divided into two groups with LVEF ≤ 50% and LVEF > 50%, log-rank = 9.048, *p* = 0.003 (Figure 3C); divided into three groups according to AF type log-rank = 9.256, *p* = 0.01 (Figure 3D); and divided into two groups, with CHA_2_DS_2_-VASc score 0-1 point and CHA_2_DS_2_-VASc score 2–5 point, log-rank = 6.232, *p* = 0.013 (Figure 3E).

### 3.4. Catheter Ablation Complications

Of the 209 total procedures, there were 18 ablation complications (8.6%), of which 7 (3.3%) were major and 11 (5.3%) were minor (Table 4). A rupture of the mitral valve chordae was treated with open heart surgery and mitral valvuloplasty. Vascular surgery was performed in one case with jugular vein subcutaneous hematoma. Two cardiac tamponades were cured with pericardiocentesis while pneumothorax was resolved with pleural drainage. The other 13 complications (6.2%) were managed conservatively.

## 4. Discussion

To the best of our knowledge, this is the first study that presents and uses a semiquantitative assessment for PVI success, which we call pulmonary vein isolation outcome degree. This original clinical trial has a novel approach in PVI outcome evaluation with a scoring of PVIOD from 1 to 4. Until now, trials have normally used a qualitative measure of PVI success with two possible outcomes: effective or unsuccessful procedure [3,5,7]. In many studies, the conventional prognostic assessment after ablation focused only on the presence or absence of AF recurrence and did not assess partial clinical success, such as reduced AF burden. A few trials reported that clinical success could be significant on top of procedural efficacy after PVI [2,4].

The importance of this trial is that PVIOD is a new PV ablation outcome scoring system. This classification considers the number and efficacy of PVI, clinical success, optional ASM and antiarrhythmic therapy. PVIOD 1 describes a successful, single PVI; PVIOD 2 refers to efficacy after multiple procedures (≥2 re-PV isolation and/or ASM); PVIOD 3 is clinical success after PVI ± ASM; and PVIOD 4 is procedural and clinical failure.

The major result of our study is that, to date, PVIOD 1–4 offers the most exact long-term prognosis of PVI and can be determined with AF type, LA diameter and LVEF. These risk factors are in significant independent association with PVIOD 1–4. Our patients with longstanding persistent AF had a 3.5-fold higher chance for each higher degree of PVIOD. In addition, with an increase in LA diameter of 1 mm, there was a 20% higher chance for each higher degree of PVIOD. With a decrease in LVEF of 1%, there was a 10% higher chance for each higher degree of PVIOD. Regarding the facts mentioned above, PVIOD adds further quantitative information. Several researchers have reported that these parameters were associated with ablation success rate, but never provided these results [2,3,4,5,6,7,8,9,10,11,12]. The type of AF, LA size and LVEF are convenient measures to predict future PVI outcomes and easy to be measure at baseline using cardiac echocardiography and clinical evaluation of patients.

After a 7-year follow-up, a single PVI had a modest success of 32.5% in our study group, with the best result of 40.2% in patients with paroxysmal AF, followed by 20.8% in persistent AF and dropping as low as 12.5% in longstanding persistent AF. Numerous trials have detected long-term success after single catheter ablation from 29% to 57% [2,4,5,7]. Repeated ablations without ASM, after the 7-year follow-up, increased cumulative success to 57.3% in our study. Additional substrate strategies, with an efficacy rate of 4.3% in our patients, generated very little benefit to procedural success, which increased to 61.6%. The work of Ouyang et al. showed that patients with paroxysmal AF and normal LVEF after single PVI (4.8-year follow-up) had a success rate of 46.6%; this was 73.9% after the second procedure and, after the third ablation, the efficacy was 79.5% [4]. Our results were excellent for patients with paroxysmal AF; after the first PVI and the 7-year follow-up, the success rate was satisfactory (40.2%), while after the last ablation, procedural success was high (68.8%) and procedural with clinical success was impressive (84.4%). Long-term clinical success was reported in 14.5% of our subjects, which significantly increased cumulative success to 76.1%, with excellent results in paroxysmal AF (84.4%), a good outcome in persistent AF (70.8%) and modest efficacy in longstanding persistent AF (43.7%). These data show the importance of clinical improvement and the role of AAD therapy in AF management. We concluded that long-term clinical success was a very significant outcome. Teunissen et al. found that clinical improvement on or of AAD was 25% and cumulative success increased to 87.5% in their total population of 509 study patients [2].

The most challenging issue in managing patients with AF is identifying those predictors for unsuccessful ablation. A large number of studies have determined risk factors for poor outcome following PVI, such as nonparoxysmal AF (particularly longstanding persistent AF) [2], LA dilatation [9], low LVEF [13] and a high CHA_2_DS_2_-VASc score [8], which were confirmed in our trial. In our study, by using nomograms with these risk factors, we could calculate both long-term procedural failure and procedural with clinical failure. Enlarged LA diameter is a well-known risk factor for PVI failure, associated with atrial electroanatomical remodeling, which plays the main role in the perpetuation and progression of AF [12,15]. In our patients, an LA diameter > 41 mm was a high predictor of procedural failure (AUC 0.726) and procedural with clinical failure (AUC 0.741). Our subjects undergoing AF repeat ablations with an LA size ≤ 41 mm had significantly better chances of survival free of AF recurrence than those with an LA size > 41 mm. The computational model confirmed that a critical LA effective conducting size > 40 mm was required for sustained multiple wavelet reentry [12]. A meta-analysis by D’Ascenzo et al., which included 7217 subjects, showed that persistent AF, LA diameter > 50 mm and arrhythmia recidivism within the first month after catheter ablation are the most powerful predictors of procedural failure [15,16]. A meta-analysis by Zhuang et al. showed an association between LA size and AF recurrence after single circumferential PVI [17]. Beukema et al. reported that 6 months after PVI, maintenance of the sinus rhythm led to atrial reverse remodeling; LA diameter decreased from 44.0 ± 5.8 mm to 40.0 ± 4.5 mm and AF recurrence increased LA diameter from 45.0 ± 6.5 mm to 49.0 ± 5.4 mm [18]. Recent studies confirmed that patients with AF recidivism following PVI had significantly higher LA volume compared to those without recurrence [19,20]. The results of Hauser et al. demonstrated that the determination of pulmonary vein size prior to catheter ablation predicts procedural outcome [21].

The most important factor that defines either PVIOD 1, 2 or 3 can be PV durability [14]. The LA diameter, as an anatomical factor, seems to be related to PV durability. Recently, PVI using a contact-force-sensing catheter or cryoballoon-based PVI has shown high PV durability and contribution to improved prognosis. When the patients included in this study underwent PVI, the skill and experience of the operators largely defined PV durability in a single procedure, which is significantly different from the prognosis prediction in the current era. However, the unpredictable resumption of electrical conduction in ablated tissue remains a source of frustration for cardiac electrophysiologists. The most important reason for recurrent AF and new atypical AFL is PV reconnection, which, in many cases, can be observed during the actual procedure, despite initial success with catheter ablation.

The CHA_2_DS_2_-VASc score is designed for thromboembolic risk assessment [11]. Our study determined that a CHA_2_DS_2_-VASc score ≥ 2 was a predictor of poor PVI outcome. Patients with CHA_2_DS_2_-VASc scores of 0 and 1 displayed successful ablations, but other scores showed high predictive value for procedural failure (AUC 0.711) and procedural with clinical failure (AUC 0.718). A score of ≥ 2 for both CHADS_2_ and CHA_2_DS_2_-VASc had the highest predictive value for AF recurrence after single catheter ablation for paroxysmal AF in the study of Letsas et al. [22]. In a relatively small cohort of patients with longstanding persistent AF, it was shown that a CHA_2_DS_2_-VASc score ≥ 3 and renal dysfunction were significantly associated with ablation failure within 31 months [23]. In the study by Kornej et al., CHA_2_DS_2_-VASc and R_2_CHADS_2_ scores were associated with PVI outcome within the first 12 months in 2069 patients with paroxysmal and persistent AF [8]. In the same trial, persistent AF, early AF recurrences and higher LA diameter were significant predictors of late AF recurrences [8]. Other scores, such as APPLE, DR-FLASH and MB-LATER, predicted electroanatomical substrate and arrhythmia recurrences in patients with AF who underwent catheter ablation [10,24].

In the current study, we identified longstanding persistent AF with paroxysmal AF as a reference category for prediction of PVI failure after a 7-year follow-up. In the relevant literature, AF type is a proven prognostic factor of PVI outcome [11,25]. Paroxysmal AF was associated with significantly higher PVI success than persistent AF in the study by Berkowitsch et al. [26]. LA remodeling due to nonparoxysmal AF may produce new triggers for AF, in addition to the PV. The prevalence and incidence of paroxysmal AF in population-based trials must have been underestimated, as arrhythmia cannot always be detected at the time of health check-ups.

The strict relationship between heart failure and AF is well recognized [11,27]. According to the guidelines, an LVEF less than 35% or longstanding persistent AF are Class IIb indications for PVI [11,13,27,28]. Many clinical studies have consistently shown the superiority of PVI over pharmacological therapy for the control of AF and, consequently, prevention of LVEF deterioration in heart failure patients [27]. Our patients with an LVEF > 50% had significantly better AF-free long-term survival after PVI than those with an LVEF ≤ 50%.

The UV ordinal logistic regression analysis showed an association of PVIOD 1–4 with structural heart disease and diabetes mellitus, but not in the MV model. Structural heart disease is a well-known predictor of early and very late recurrence of AF after RF ablation [9]. Cardiac risk factors, such as hypertension and diabetes mellitus, are closely correlated with inflammation and, consequently, with atrial fibrosis and remodeling [28]. Although it did not reach significance in our MV analysis, the importance of diabetes mellitus prevention and treatment in AF patients is obvious.

A multidisciplinary approach in the management of patients with AF is essential because of the necessity for polypharmacy, the presence of comorbidities and the demand for continuous complex technological advances in RF and cryoballoon isolation of PV. In the current guidelines, recommendations for PVI as selection criteria include the presence of symptoms, AF type and LVEF [13]. Many prognostic factors of PVI outcome, such as LA size, CHA_2_DS_2_-VASc score and patient age, should be included in novel guidelines for catheter ablation of PV. As AAD is commonly associated with serious side effects, catheter ablation can be used much more often as a first-line strategy in AF treatment [13]. However, despite the tremendous difficulties, the realization of new clinical scores and guidelines for PVI should be a goal in the future.

This study’s results show significant major ablation complications in 3.3% of procedures, which is acceptable and can compare with the results (2.5–8%) obtained in other eminent electrophysiology laboratories [29,30]. There were no deaths connected to catheter ablations among our patients, which is an excellent result. According to these findings and the data from the literature, we concluded that PVI is a safe procedure that is evolving and shows promising success in AF treatment.

This study is limited because of the relatively small sample size and its nature as a single-center trial. PVIOD needs to be tested in future studies with a more homogenous and larger group of patients, particularly to confirm the influence of diabetes mellitus and structural heart diseases in the genesis of atrial fibrillation. Another limitation of this study is that the female sex had a low representation (21.5%). An explanation for this may be that we used consecutive patients for our trial, regardless of sex. In the study by Teunissen et al., the male sex was predominant (75.8%), similar to our trial (79.5%) [2]. In addition, the results obtained by Ouyang also showed male predominance (75.2%) in patients undergoing AF catheter ablation [4]. In the last few years’ of studies, LA volume has been preferred to LA diameter as a predictor of PVI outcome [19,20]. Our study protocol was written in 2012 when LA diameter was a standard, widely used prognostic tool, available and easily measured in all patients who underwent PVI.

## 5. Conclusions

This study developed the pulmonary vein isolation outcome degree as a new semi-quantitative measure for PVI success. To date, PVIOD 1–4 is the most exact long-term prognosis of PVI and can be determined based on knowledge of AF type, LA diameter and LVEF. Patients with longstanding persistent AF had 3.5-fold higher chance of developing a higher degree of PVIOD. In addition, with an LA diameter increase of 1 mm, the chances of a higher degree of PVIOD rose by 20%. An LVEF decrease of 1% increased the chance of a higher degree of PVIOD by 10%. An LA size greater than 41 mm, an LVEF ≤ 50% and longstanding persistent AF were strong significant predictors of 7-year PVI failure and identified patients who were likely to experience AF recurrence. We hope that the PVIOD 1–4, as a novel score, will be useful not only for clinical practice but also for the development of prognostic methods in the medical and biological fields.

## Figures and Tables

**Figure 1 jcm-10-05827-f001:**
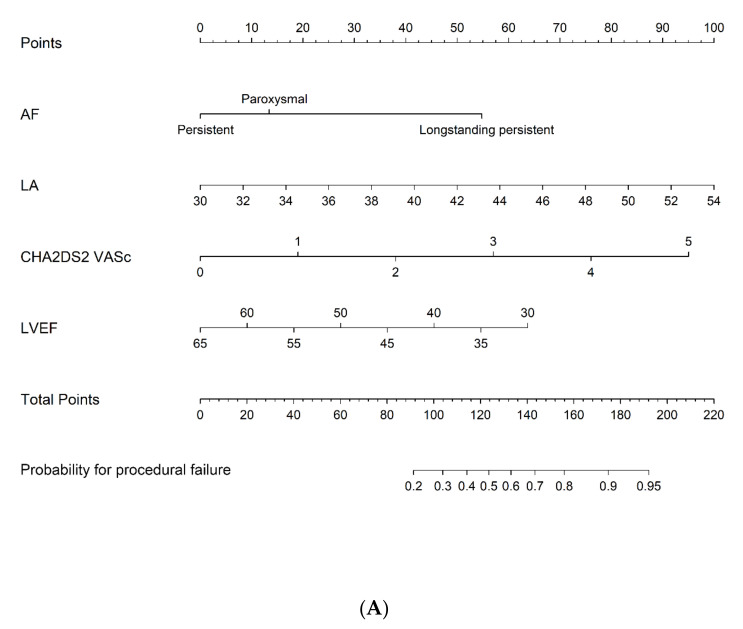
Nomogram 1 shows the influence of AF type, LA diameter, LVEF and CHA_2_DS_2_-VASc score on the 7-year probability for procedural failure (**A**) and nomogram 2 calculates the 7-year probability for procedural and clinical failure (**B**). AF: atrial fibrillation (type: a. paroxysmal; b. persistent; c. longstanding persistent); LA: left atrial diameter; LVEF: left ventricular ejection fraction; CHA_2_DS_2_-VASc: CHA_2_DS_2_-VASc score.

**Figure 2 jcm-10-05827-f002:**
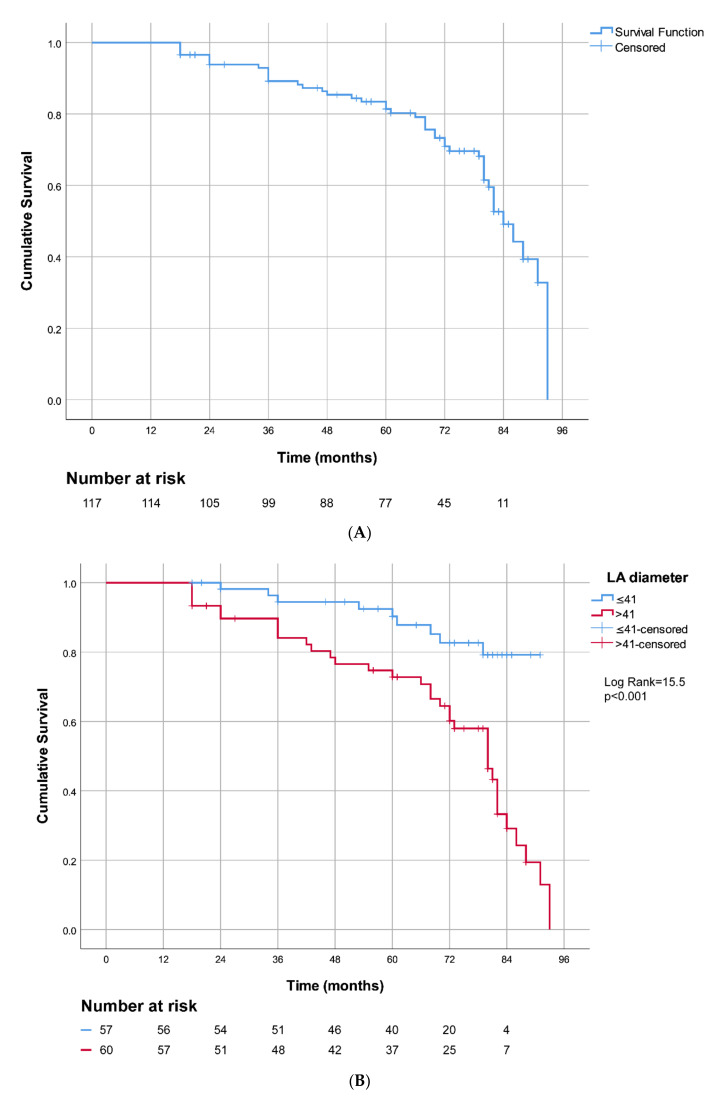
Kaplan–Meier curves for the AF-free survival after PVI in patients compared procedural success and procedural failure (**A**). Kaplan–Meier curves in patients with LA diameter ≤ 41 mm and LA diameter > 41 mm (**B**). LVEF ≤ 50% and LVEF > 50% (**C**). Three types of AF (**D**). CHA_2_DS_2_-VASc score of 0–1 and CHA_2_DS_2_-VASc score of 2–5 (**E**).

**Figure 3 jcm-10-05827-f003:**
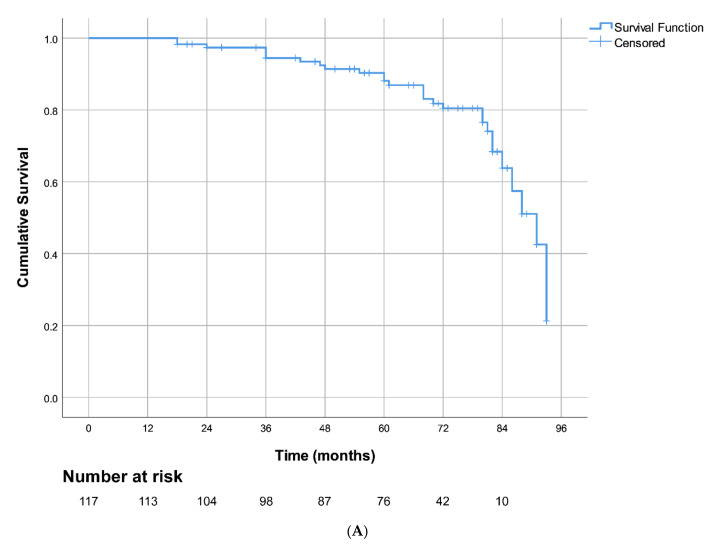
Kaplan–Meier curves for the AF-free survival after PVI in patients, comparing procedural and clinical success with procedural and clinical failure (**A**). Kaplan–Meier curves in patients with LA diameter ≤ 41 mm and LA diameter > 41 mm (**B**). LVEF ≤ 50% and LVEF > 50% (**C**). Three types of AF (**D**). CHA_2_DS_2_-VASc score of 0–1 and CHA_2_DS_2_-VASc score of 2–5 (**E**).

**Table 1 jcm-10-05827-t001:** Baseline and PVIOD characteristics of the study patients.

Parameter	All	PVIOD 1	PVIOD 2	PVIOD 3	PVIOD 4	*p*-Value
	*n* = 117	*n* = 38	*n* = 34	*n* = 17	*n* = 28	
Age (years)	56.2 ± 8.5	56.0 ± 7.8	54.3 ± 9.4	59.8 ± 8.3	56.8 ± 8.3	0.189
Sex (male)	93 (79.5%)	30 (78.9%)	30 (88.2%)	14 (82.4%)	19 (67.9%)	0.26
Duration of AF (years)	5 (1–18)	5 (1–16)	4 (1–16)	8 (1–18)	5.5 (2–15)	0.125
Paroxysmal AF	77 (65.8%)	31 (81.6%)	22 (64.7%)	12 (70.6%)	12 (42.9%)	
Persistent AF	24 (20.5%)	5 (13.2%)	10 (29.4%)	2 (11.8%)	7 (25%)	0.009
Longstanding persistent AF	16 (13.7%)	2 (5.3%)	2 (5.9%)	3 (17.6%)	9 (32.1%)	
BMI (kg/m^2^) *	27.7 ± 4.0	26.9 ± 4.3	27.8 ± 4.1	27.5 ± 3.4	28.6 ± 3.8	0.553
CHA_2_DS_2_-VASc score	1 (0-5)	1 (0-3)	0 (0–3)	1 (0–5)	2 (0–4)	<0.001
Hypertension	75 (64.1%)	24 (63.2%)	16 (47.1%)	12 (70.6%)	23 (82.1%)	0.035
Diabetes mellitus	9 (7.7%)	2 (5.3%)	1 (2.9%)	1 (5.9%)	5 (17.9%)	0.183
Hypercholesterolemia	53 (45.3%)	18 (47.4%)	12 (35.3%)	11 (64.7%)	12 (42.9%)	0.252
Structural heart disease	21 (17.9%)	5 (13.2%)	3 (8.8%)	3 (17.6%)	10 (35.7%)	0.046
Left atrial diameter (mm)	41.9 ± 4.7	39.3 ± 3.8	41.9 ± 4.6	43.1 ± 5.1	44.7 ± 3.9	<0.001
LVEF	54.8 ± 6.9	56.8 ± 4.7	57.1 ± 5.7	52.7 ± 6.2	50.7 ± 6.2	<0.001
Propafenone	33 (28.2%)	15 (39.5%)	8 (23.5%)	4 (23.5%)	6 (21.4%)	0.312
Betablocker	70 (59.8%)	23 (60.5%)	19 (55.9%)	9 (52.9%)	19 (67.9%)	0.726
Antiarrhythmic group III	68 (58.1%)	17 (44.7%)	22 (64.7%)	12 (70.6%)	17 (60.7%)	0.207
Verapamil	5 (4.3%)	3 (7.9%)	0 (0.0%)	0 (0.0%)	2 (7.1%)	0.276

Results are shown as number (percentage), mean ± standard deviation or as median (interquartile range). BMI: body mass index; * *n* = 93 patients.

**Table 2 jcm-10-05827-t002:** PVIOD in patients with paroxysmal, persistent and longstanding persistent AF.

	Paroxysmal AF *n* = 77	Persistent AF *n* = 24	Longstanding p. AF *n* = 16	All *n* = 117
PVIOD 1	40.2	20.8	12.5	32.5
PVIOD 2	28.6 (68.8)	41.7 (62.5)	12.5 (25)	29.1 (61.6)
PVIOD 3	15.6 (84.4)	8.3 (70.8)	18.7 (43.7)	14.5 (76.1)
PVIOD 4	15.6 (100)	29.2 (100)	56.3 (100)	23.9 (100)

Results are shown as percentage (%) and cumulative %. p.: persistent.

**Table 3 jcm-10-05827-t003:** Univariate and multivariate ordinal logistic regression analyses.

Predictor		UV			MV	
	OR	95% CI	*p*-Value	OR	95% CI	*p*-Value
Age (years)	1	1.0–1.1	0.43			
Sex (male)	1.5	0.7–3.5	0.289			
Duration of AF (years)	1.1	1.0–1.2	0.127			
Paroxysmal AF	Referent			Referent		
Persistent AF	1.9	0.8–4.4	0.124	1.1	0.4–2.8	0.9
Longstanding persistent AF	6.5	2.3–18.6	0.001	3.5	1.1–10.7	0.031
BMI (kg/m^2^) *	1.1	1.0–1.2	0.173			
CHA_2_DS_2_-VASc score	1.5	1.1–2.1	0.008	1.4	0.95–2.0	0.086
Hypertension	1.8	0.3–3.5	0.105			
Diabetes mellitus	3.6	1.0–13	0.049	2.7	0.6–12.1	0.205
Hypercholesterolemia	1	0.5–2.0	0.895			
Structural heart disease	3	1.2–7.2	0.014	0.4	0.1–1.6	0.21
Left atrial diameter (mm)	1.2	1.1–1.3	<0.001	1.2	1.1–1.3	0.001
LVEF	0.9	0.85–0.95	<0.001	0.9	0.86–1.0	0.038
Propafenone	0.5	0.2–1.1	0.086			
Betablocker	1.1	0.6–2.2	0.695			
Antiarrhythmic group III	1.7	0.9–3.4	0.109			
Verapamil	0.6	0.1–3.1	0.545			

**Table 4 jcm-10-05827-t004:** Complications in 209 catheter ablations.

Complications	Number (%) *n* = 18 (8.6%)
Major:	7 (3.3%)
Cardiac tamponade	2
Rupture of mitral valve chordae	1
Pneumothorax	1
Stroke	1
Retroperitoneal hematoma	1
Jugular vein subcutaneous hematoma	1
Minor:	11 (5.3%)
Pericardial effusion	5
Inguinal subcutaneous hematoma	5
Superficial thrombophlebitis	1

## Data Availability

This publication has been read and approved by all authors, as well as by the responsible authorities at the Institute for Cardiovascular Diseases Dedinje, where the work was carried out. The de-identified participant data will be shared on a request basis. Requests for data sharing can be directed to the author. The statistical analysis plan, data base with all patients’ data—clinical, echocardiography, radiofrequency ablations characteristics, etc. will be shared by the authors. The data will be available immediately after the publication and will be shared with anyone on a request basis; data, along with explanations, can be sent via e-mail. The publication of this manuscript implies that the authors must make all materials, data and protocols associated with the publication available to readers.

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
