# Peer review of "Pulmonary Vein Isolation Outcome Degree Is a New Score for Efficacy of Atrial Fibrillation Catheter Ablation"

_jcm, 2021, doi:10.3390/jcm10245827_

Round 1
Reviewer 1 Report
Jurcevic et al. presented a manuscript in which they introduced a Pulmonary Vein Isolation Outcome Degree (PVIOD) as a new measure for AF cather ablation success. They included 117 patients who underwent to Pulmonary vein isolation and they followed these patients for 7 years. The authors recorded baseline characteristics including Left atrial diameter, CHA2DS2VASc score, ventricular function, type of AF.
The study is clear, well-written.
Here are my comments:
- The authors should explain in their discussion the reason why you used 41 mm cut-off for LAd. I guess it is for the male predominace in this study (almost 80%). Since they showed how LAd >41 mm is associated to failure to AF ablation, this could be not true for female patients since cut-off used for LAd is 39mm.
- Moreover, female sex has a low representation and this limitation could be mentioned.
- Kaplan Meier survival curves should be presented with log-rank and p-value (figure 3B, 3C, 3D, 3E and 4B, 4C, 4D, 4E).
Author Response
Dear Reviewer,
Thank You very much for suggestions You have made about my article, which improved it. I send You response to comments in two files: "Cover letter" and "Response to Reviewer 1 ".
Please see the attachment.

Reviewer 2 Report
The authors created new assessment score for success of AF ablation; pulmonary vein isolation outcome degree (PVIOD). PVIOD was associated with long-standing persistetnt AF, larger LAD, lower LVEF in univariate analysis. PVIOD and large LAD were multivariate predictors for AF ablation outcome. I have some concerns about this paper.
- The authors should mention the definition of grouping more clearly. I had trouble interpreting this paper. This issue might be the most important if the authors revise this paper. It was difficult to grasp the outline reading this abstract.
- The long follow-up duration of 7 years was the advantage of this paper. But, I think the number of patients was not large enough to perform these analysis.
- Although I agreed with the shown results partially, the similar analysis has been performed in the previous papers. Is LAD more useful than PVIOD?
- The current method of ASM may be different from this paper.
Author Response
Dear Reviewer,
Thank You very much for suggestions You have made about my article which improved it. I send You response to comments in file " Response to Reviewer 2'.
Please see the attachment.

Reviewer 3 Report
The authors investigated whether PVI outcome degree (PVIOD) works as a semi-quantitative measure for the success of AF ablation. Using multivariate logistic regression analysis, PVIOD was associated with LS-peaf, LA diameter, LVEF. Consequently, the authors concluded PVIOD is a novel AF ablation outcome scoring system.
The study is straightforward in its retrospective design. The results seem intuitive as the type of AF, LA diameter, or ejection fraction are well-established risk factors for recurrence of atrial arrhythmia following AF ablation. I have the following comments:
Major:
- The description regarding the PVIOD in the introduction seemed to be abrupt. There would be room for improvement in this paragraph.
- As the relatively small number of subjects (n=117) in this study, the statistical validity would be further lacking due to dividing 4 groups (PVIOD1-4). Please clarify the rationale for the classification.
- The definition of recurrence after ablation (clinical success) is not understandable. Please describe the details of the definition in atrial arrhythmia reduction, especially. Further, the data on AADs after the blanking period are lacking. Please clarify.
Minor:
- The reference cited in the methods about the classification of AF type was not up-to-date.
- Please clarify the median follow-up days after ablation.
- “Number at risk “in the KM curves is lacking.
- Please try not to repeat data in the tables again in the text.
- There are too many figures and tables. Both the paper, especially in the discussion, and illustrations could be shortened substantially.
Author Response
Dear Reviewer,
Thank You very much for suggestions You have made about my article which improved it. I sent You response to comments in file "Response to Reviewer 3".
Please see the attachment.

Round 2
Reviewer 2 Report
In the revised manuscript, the authors clearly described the definition of pulmonary vein isolation outcome degree (PVIOD), and sophisticated the entire manuscript. The authors analyzed the association between PVIOD and several clinical variables. However, I still have some fundamental concerns about the objective of this paper.
I wonder what the authors attempted to demonstrate in this paper. I can grasp the definition of PVIOD in the revised version. The authors stratified the outcome after AF ablation by procedure and clinical success. I agree in that PVIOD was associated with type of AF, LA dilatation and LV systolic dysfunction. However, several researchers have been reported that these parameters were associated with poor ablation success rate. Does the information of PVIOD add further information? PVIOD is not a predictor of ablation outcome but outcome itself.
Author Response
Dear Reviewer,
Thank you very much for the suggestions you made concerning our manuscript. Thanks to your comments, we made new conclusions and improved the rationale for introducing PVIOD in this study.
We hope that you will be satisfied with explanations and changes in the paper that we have accepted based on your comments.
We agree with your suggestion that PVIOD represents an ablation outcome itself. According to that, we made changes in all sections of this manuscript: abstract, introduction, methods, results, discussion and conclusions.
The purpose of the present article is to expand the quantitative measure of procedural success in the medical and biological fields. In this paper we have demonstrated that PVIOD 1-4 was the most exact long-term prognosis of PVI and that it could be determined using AF type, LA diameter and LVEF. For the first time, in our trial, we confirmed that these 3 risk factors are in significant independent association with PVIOD 1-4. Our patients with longstanding persistent AF had 3.5-time higher chance for developing higher degree of PVIOD. Also, with LA diameter increased for 1 mm, rising chances for a higher degree of PVIOD are 20%. LVEF decrease of 1% increases the chance for a higher degree of PVIOD for 10%. Several researchers have reported that these parameters were associated with ablation success rate, but never provided quantified outcome. Until now the trials usually used a qualitative measure of PVI success with 2 possible outcomes: effective or unsuccessful procedure. Teunissen et al. suggested that long-term clinical success could be very important on top of procedural success after AF catheter ablation [2]. Regarding the above mentioned, PVIOD 1-4 add further quantitative information. We hope that the PVIOD 1-4 as a new score will be useful for not only clinical practice, but also for the development of prognostic methods in medical and biological fields.
Regarding the comments provided in English language, English language in this final version of our manuscript has been edited by Akademija Oxford agency, Belgrade.
We hope that you will accept this improved final version of the manuscript for publishing in the Journal of Clinical Medicine.
Best regards,
MSc Ruzica Jurcevic,
Institute for Cardiovascular Diseases Dedinje
E-mail: ruzicajurcevic@hotmail.com
Reviewer 3 Report
The authors responded to the comments. I have no further questions or comments.
Author Response
Dear Reviewer,
Thank you very much for the suggestions you made concerning our manuscript. We hope that you will be satisfied with explanations and changes in the paper that we have accepted based on your comments. English language and style in this final version of our article has been edited by Akademija Oxford agency, Belgrade.
Best regards,
MSc Ruzica Jurcevic,
Institute for Cardiovascular Diseases Dedinje
E-mail: ruzicajurcevic@hotmail.com
